# Influence of preoperative frailty on quality of life after cardiac surgery: Protocol for a systematic review and meta-analysis

Kathryn Bezzina[1], Christophe A. Fehlmann[2,3,4]*, Ming Hao Guo[5], Sarah M. Visintini[6], Fraser D. Rubens[5], George A. Wells[2,7], Rosetta Mazzola[2], Caroline McGuinty[8], Allen Huang[1,3], Lara Khoury[1], Kevin Emery Boczar[2,8]

1 Division of Geriatric Medicine, The Ottawa Hospital, Ottawa, Ontario, Canada, 2 School of Epidemiology and Public Health, University of Ottawa, Ottawa, Ontario, Canada, 3 Ottawa Hospital Research Institute, Ottawa, Ontario, Canada, 4 Division of Emergency Medicine, Geneva University Hospitals, Geneva, Switzerland, 5 Department of Cardiac Surgery, University of Ottawa Heart Institute, Ottawa, Ontario, 6 Berkman Library, University of Ottawa Heart Institute, Ottawa, Ontario, Canada, 7 Research Methods Centre, University of Ottawa Heart Institute, Ottawa, ON, Canada, 8 University of Ottawa Heart Institute, Ottawa, Ontario, Canada

☯ These authors contributed equally to this work.
* christophe.fehlmann@hcuge.ch

**Funding:** The author(s) received no specific funding for this work.

**Competing interests:** The authors have declared that no competing interests exist.

## Abstract

### Background

Frailty has emerged as an important prognostic marker of adverse outcomes after cardiac surgery, but evidence regarding its ability to predict quality of life after cardiac surgery is currently lacking. Whether frail patients derive the same quality of life benefit after cardiac surgery as patients without frailty remains unclear.

### Methods

This systematic review will include interventional studies (RCT and others) and observational studies evaluating the effect of preoperative frailty on quality-of-life outcomes after cardiac surgery amongst patients 65 years and older. Studies will be retrieved from major databases including the Cochrane Central Register of Controlled Trials, Embase, and Medline. The primary exposure will be frailty status, independent of the tool used. The primary outcome will be change in quality of life, independent of the tool used. Secondary outcomes will include readmission during the year following the index intervention, discharge to a long-term care facility and living in a long-term care facility at one year. Screening, inclusion, data extraction and quality assessment will be performed independently by two reviewers. Meta-analysis based on the random-effects model will be conducted to compare the outcomes between frail and non-frail patients. The evidential quality of the findings will be assessed with the GRADE profiler.

### Conclusion

The findings of this systematic review will be important to clinicians, patients and health policy-makers regarding the use of preoperative frailty as a screening and assessment tool before cardiac surgery.

**Study registration**

OSF registries (https://osf.io/vm2p8).

## Introduction

As the life expectancy of the population has increased, the prevalence of coronary heart disease has also risen [1]. Thus, with these trends, the number of referrals for cardiac procedures for older adults continues to rise. However, there are unique characteristics in this aging population that must be considered when contemplating referral for cardiac surgery. In particular, frailty needs to be assessed and considered when pursuing invasive cardiac interventions. Frailty is a construct that describes an increased vulnerability of people to minor stressors, due to diminishing system reserves and has an increased prevalence with aging [1]. However, "biological age" is often mistakenly taken as a surrogate marker of frailty, where in reality, upwards of 75% of patients over the age of 85 years are not frail [2].

The importance of frailty has been acknowledged by the American Heart Association (AHA) which has called for a better understanding of its role in cardiac care [3]. There has been a call for trials to better improve the characterization of an older adult population and to evaluate the role of people living with frailty on clinical outcomes. However, frailty can be extremely difficult to quantify. While frailty does not yet have an internationally accepted definition, in general it is thought to be a heterogeneous syndrome that reflects multi-system dysfunction in which individuals are able to freely transition between severity states [4–8]. Although it is more commonly encountered in an ageing population, it is a distinct entity from ageing. Multiple validated scores have been developed to attempt to operationally define frailty in clinical practice. These include, but are not limited to, the Fried score, the Clinical Frailty Scale (CFS) and the Edmonton Frailty Scale (EFS) [6, 9, 10]. However, there is no single accepted score that is universally used in clinical practice. In general, an assessment tool for frailty should be able to accurately identify frailty, reliably predict adverse clinical outcomes, predict response to potential therapies, be supported by biological causative theory, and be simple to apply [4, 7, 11, 12].

Currently, more than half of cardiac surgery procedures are being performed on patients aged 75 years and older [13]. As this population continues to grow rapidly, it has never been more important to understand the impact that the presence of frailty can play on the hospital course, recovery and overall prognosis of these patients.

A comprehensive preoperative assessment is essential to determine a patient's preoperative relative risk and to determine the degree of relative benefit they could possibly derive from a procedure. The Society of Thoracic Surgeons (STS) risk score and The European System for Cardiac Operative Risk Evaluation (EuroSCORE) II are the most widely used tools to evaluate perioperative mortality and morbidity in patients undergoing cardiac surgery, and the STS is considered the global standard for this purpose [14–17]. However neither include comprehensive evaluations for frailty [18]. Thus, there has been an increased focus on the utility of a dedicated preoperative frailty assessment for patients undergoing cardiac surgery in recent years.

Importantly, while the assessment of frailty has been shown to predict perioperative mortality and morbidity, there is a paucity of literature on its implications for long-term functional outcomes. From a patient-perspective, when considering therapeutic treatment options in older adults, the outcome of quality of life following an intervention can be even more important to the patient than the outcome of mortality. Yet mortality is more commonly investigated

in clinical trials [19]. Thus, research regarding patient-centred outcomes related to frailty and cardiac surgery is greatly needed.

## Objectives

This study aims to review the frailty assessment instruments being used for cardiac surgery patients and to assess the prognostic utility of preoperative identification of frailty in determining quality of life post-cardiac surgery. The specific review questions will include: (1) how is frailty being assessed in patients undergoing cardiac surgery? and (2) how does quality of life change after cardiac surgery in people living with frailty?.

## Methods

This protocol follows the PRISMA-P guidelines [20]. The final report will be guided by the PRISMA Statement for Reporting Systematic Reviews and Meta-Analyses [21]. The checklist can be found in S1 Appendix. The protocol has been registered on OSF registries, the Open registries network (https://osf.io/vm2p8).

### Population

We will include studies examining adults 65 years and older who underwent cardiac surgery. Cardiac surgeries that will be considered are coronary artery bypass grafting, valvular surgery and transcatheter valve interventions (transcatheter aortic valvular replacement (TAVR), thoracic aorta surgeries, and mitral valve repair (TEER). Left Ventricular Assist Device (LVAD) and heart transplantation procedures were not included, as it was felt these procedures were fairly unique in terms of their procedural associated risk factors, and were also not particularly relevant for our population of interest.

### Frailty assessments

To be eligible, studies should report baseline frailty assessments, independent of the tool used [4, 6, 22, 23]. Studies reporting frailty assessed only by biological (single or multiple) or radiological finding (i.e. sarcopenia on computed tomography [CT]), single systems or clinical judgement alone will be excluded. Thus, only studies that utilize a comprehensive frailty assessment which is multifaceted and considers multiple physiological, physical, cognitive, social and/or functional domains will be included.

As many frailty assessment tools do not have a specific cut-off or threshold for defining frailty, we will exclude studies that fail to dichotomize the study population into frail and non-frail study groups. We will also exclude those studies in which the frailty cut-off was defined by the study population under investigation (i.e. if a frailty cut-off was defined by a percentile or median). If different studies use the same method for assessing frailty but use different cut-off scores for defining frailty, we will utilize the threshold that was defined by each study in question.

### Outcomes

The primary outcome will be change in quality of life between the time preceding and following the surgery, independent of the tool used to measure quality of life. Secondary outcomes include readmission during the year following the index intervention, discharge to a long-term care facility and living in a long-term care facility at one year.

## Information sources

We will search Medline, Embase, and the Cochrane Central Register for Clinical Trial from inception until 04/01/2021. Additionally, a manual search of all eligible articles' reference lists, articles citing eligible articles as well as relevant review articles will be carried out in order to identify any additional literature.

## Search strategy

The search strategy will be developed by a medical librarian, then peer-reviewed by a second librarian as per PRESS guidelines [24]. Search terms related to cardiac surgery, frailty, and older patients will be included. S2 Appendix shows a draft of the search strategy.

## Eligibility criteria

We will include studies reporting original research (randomized controlled trials, non-randomized interventional studies, prospective or retrospective comparatives cohorts and case-control studies), regardless of the number of included patients. Letters, editorials, review articles, case reports and case series ($\leq$10 patients) will be excluded. Conference abstracts from the previous year will be screened to identify additional manuscripts not found with the initial search. Studies will only be eligible if they report on cardiac surgery in adult patients $\geq$ 65 years old. If studies reported mixed surgeries, authors will be contacted to attain stratified data for patients with cardiac surgery. If stratified data are not available, studies will not be included. Articles reported in languages other than English will be excluded. As the focus of this review is on frailty, rather than preoperative baseline surgical risk, we will not use baseline surgical risk as an exclusion criterion.

## Selection process

The results of the literature search will be uploaded to Covidence Software and all included articles will be allocated a unique study identification code to track articles throughout the data screening and extraction process [25]. Duplicates will be removed either electronically during the search or manually during screening. Covidence software will be used for level 1 (Title and abstract) and level 2 (Full text) assessment. Titles and abstracts yielded by the search will be screened independently by two reviewers. Discrepancies will be resolved by a third reviewer. Full reports of titles and abstracts that meet inclusion criteria will be reviewed independently by two reviewers. Discrepancies will also be resolved by the other reviewer. Reasons for excluding full text articles will be recorded. If two or more papers report the results for the same outcome in the same study, only the study with the larger sample size will be chosen. Study selection will be documented and summarized in a PRISMA-compliant flow-chart.

## Data collection process

For data extraction, a pre-designed, standardized data extraction sheet will be used. The reviewers will first test the extraction sheet on two studies. They will then discuss and make amendments if necessary. Finally, two reviewers will independently collect the prespecified data. Disagreements will be resolved by the third reviewer. If necessary, study authors will be contacted by email for more information. If no replies are received, two subsequent emails will be sent at 2 and 4 weeks.

## Data items

For each study, we will collect publication details (author, year of publication, country, journal), study details (study design, eligibility criteria, number of patients included, funding resource, author conflicts of interest), patients characteristics (age, sex, surgery type, emergency or elective intervention), frailty measure (type of measure, chosen cut-off), and sample size of frail and non-frail. The pre-specified outcomes will be extracted according to frail and non-frail for each group, in each study. If essential data are not reported, authors will be contacted for more information.

## Risk of bias in individual studies

Most of the studies will likely be trial and cohort studies. Risk of bias will then be evaluated based on the Newcastle-Ottawa Scale (NOS) or Cochrane Risk of Bias tool (RoB), depending on the type of study [26, 27]. If there is insufficient detail reported, we will judge the risk of bias as 'unclear' and try to contact the study investigators for additional information. Bias will be evaluated independently, and any discrepancies will be resolved through discussion until consensus is reached, with the assistance of a third reviewer when necessary.

## Data synthesis

Clinical heterogeneity will be evaluated based on study population, design, and assessment of the outcomes. If at least two studies are judged to be clinically homogeneous, a meta-analysis will be conducted using a random effects model.

For the primary outcome, we will calculate the mean change in quality of life along between baseline and follow-up. As it is extremely likely studies will report different measures of QOL, the standardized change score (standardized mean difference, SMD) will be calculated for each study and then be pooled using a random effects model. We will report the estimate and its 95% confidence interval. For secondary outcomes, we will pool dichotomous data and report odds-ratios and 95% confidence intervals. If, for some outcomes, there is not enough data to effectuate a meta-analysis, results will be reported descriptively.

For each meta-analysis performed, statistical heterogeneity will be evaluated through the $I^2$ statistic. If this statistic is greater than 50%, we plan to explore possible sources of heterogeneity, using subgroup and sensitivity analyses [28]. These analyses could include restriction to high-quality studies or exclusion of frailty measure reported only in one study. Finally, potential publication bias will be assessed with funnel plots. Review Manager 5.1 will be used for all statistical analyses [29].

## Confidence in cumulative evidence

We will use Grading of Recommendations Assessment, Development and Evaluation (GRADE) system [30]. The five GRADE considerations (i.e. risk of bias, consistency of effect, imprecision, indirectness and publication bias) will be used to assess the quality of the body of evidence for each outcome with a meta-analysis, and to draw conclusions about the quality of evidence within the text of the review.

## Patient and public involvement statement

Due to the nature of this study, patients are not involved in this project.

## Discussion

Frailty has been recognized as an important perioperative marker of risk in recent years [31]. A growing body of literature has shown that it is a marker of mortality in patients undergoing all types of cardiac surgery [14, 32–36]. However, whereas much attention has been focused on mortality, less emphasis has been placed on functional outcomes and quality of life [37].

Furthermore, while frailty has been shown to be an important prognostic marker of surgical risk in older patients, there is a lack of consensus about the best means to assess perioperative frailty accurately and efficiently, and the variability of its definition and assessment in the literature illustrates this point. The lack of uniformity across studies in the definition and assessment tools used for frailty makes this field of study challenging. To this end, Dent *et. al* reviewed over 12 different tools used in clinical and research domains for the assessment of frailty; the tools varied widely in their components, the time required to complete them, and their validation with respect to prediction of outcomes [4]. Additionally, they found that many studies are using modified, non-validated versions of existing tools, which can have a profound impact on frailty determination [4]. This was also seen in work by Theou *et. al*, which found 262 different versions of Fried's Frailty Phenotype [38].

Based on these findings, our goal is to help improve the assessment of frailty in older cardiac surgery patients by identifying validated tools that are being used for clinical research purposes in the area, and to determine the prognostic importance of frailty assessment on patient-centred outcomes.

Our review has several strengths. Firstly, we will perform a comprehensive search of the literature to identify published studies and will include different type of studies (RCT, cohort, case-controls) and references from previous reviews as well. It will allow us to obtain the best current evidence on this topic. Secondly, every step from the screening to the extraction and the quality assessment will be performed independently by two reviewers. Then, we plan to use the GRADE tool to evaluate the quality of evidence of the studies identified through our search. Finally, to our knowledge, this is the first study to evaluate the prognostic importance of frailty on patient-centered outcomes post-cardiac surgery and to potentially pool estimates of quality of life from multiple studies.

Our study also has important limitations. As already outlined, while many frailty tools are similar, different ones cannot be assumed to be interchangeable [39]. Although we will look at a subgroup analysis to further clarify this issue, pooling results across studies that used different methodology to assess frailty may result in significant heterogeneity. Additionally, we will likely not have the ability to look at the role that different frailty dimensions play on the outcome of quality of life for patients post-cardiac surgery. Unlike studies evaluating mortality or cardiac events, looking at quality of life can be more nuanced and fraught with difficulty. There are also multiple scales used to assess quality of life. In this regard, we may also have problems pooling our results from different studies where different methods were used to assess post-surgical quality of life. We will again try to analyse this through a subgroup analysis, but this may be subject to error. If studies use multiple indices to assess frailty, we will only use one for our meta-analysis; this is a source of potential selection bias.

With an increasing number of older adult patients undergoing cardiac surgery, the goal of our study to evaluate how frailty impacts quality of life post-procedure is of high clinical value. While preoperative interventions that help improve outcomes for frail patients are still ongoing, preliminary work has shown that there may be positive impacts on surgical outcomes for these patients [40, 41].

The results of this study can inform clinicians, patients and policy decision-makers with respect to the best clinical evidence available about the role that frailty plays on patient-centred

outcomes post-cardiac surgery. We anticipate our findings will help to fill a knowledge gap in the field and spur potential new research in the area of frailty and cardiac surgery.

## Conclusion

While frailty has been identified as a predictor of morbidity and mortality following cardiac surgery, less is known about its effects on quality of life after surgery. As an increasing number of older people living with frailty present to cardiac surgery, the importance of understanding the impact of frailty on quality of life outcomes becomes increasingly important to help inform shared clinician-patient decision-making surrounding treatment decisions and to maximize the benefits of advanced cardiac interventions.

## Supporting information

**S1 Appendix. Appendix is the PRISMA-P checklist.**
(DOC)

**S2 Appendix. Appendix is a draft of the search strategy.**
(DOCX)

## Author Contributions

**Conceptualization:** Kathryn Bezzina, Christophe A. Fehlmann, Ming Hao Guo, Kevin Emery Boczar.

**Investigation:** Sarah M. Visintini.

**Methodology:** Kathryn Bezzina, Christophe A. Fehlmann, Ming Hao Guo, Sarah M. Visintini, George A. Wells, Kevin Emery Boczar.

**Supervision:** George A. Wells, Allen Huang, Lara Khoury.

**Writing – original draft:** Kathryn Bezzina, Christophe A. Fehlmann, Kevin Emery Boczar.

**Writing – review & editing:** Kathryn Bezzina, Christophe A. Fehlmann, Ming Hao Guo, Sarah M. Visintini, Fraser D. Rubens, George A. Wells, Rosetta Mazzola, Caroline McGuinty, Allen Huang, Lara Khoury, Kevin Emery Boczar.

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
