## [Decision Letter · Decision Letter 0]

23 Sep 2021

PONE-D-21-10086

Influence of preoperative frailty on quality of life after cardiac surgery: protocol for a systematic review and meta-analysis

PLOS ONE

Dear Dr. Fehlmann,

Thank you for submitting your manuscript to PLOS ONE. After careful consideration, we feel that it has merit but does not fully meet PLOS ONE’s publication criteria as it currently stands. Therefore, we invite you to submit a revised version of the manuscript that addresses the points raised during the review process.

We look forward to receiving your revised manuscript.

Kind regards,

Laura Pasin

Academic Editor

PLOS ONE

Journal Requirements:

[CAFis funded by a Geneva University Hospital fellowship stipend. KEB is supported by a CIHR Fellowship Award.]

 [The author(s) received no specific funding for this work.]

Reviewers' comments:

Reviewer's Responses to Questions

**Comments to the Author**

1. Does the manuscript provide a valid rationale for the proposed study, with clearly identified and justified research questions?

Reviewer #1: Yes

Reviewer #2: Partly

2. Is the protocol technically sound and planned in a manner that will lead to a meaningful outcome and allow testing the stated hypotheses?

Reviewer #1: Yes

Reviewer #2: Partly

3. Is the methodology feasible and described in sufficient detail to allow the work to be replicable?

Reviewer #1: Yes

Reviewer #2: No

4. Have the authors described where all data underlying the findings will be made available when the study is complete?

Reviewer #1: Yes

Reviewer #2: Yes

5. Is the manuscript presented in an intelligible fashion and written in standard English?

Reviewer #1: Yes

Reviewer #2: Yes

6. Review Comments to the Author

You may also provide optional suggestions and comments to authors that they might find helpful in planning their study.

Reviewer #1: My suggestion is to try to find or establish an homogeneous definition of fragility in order to reduce bias

Reviewer #2: The authors performed a protocol for a systematic review and meta-analysis about frailty on quality of life after cardiac surgery.

the argument is potentially interesting but the manuscript is not to easy to read and it doesn't impact the clinical at the moment.

- page 5 line 106: why only 65 yo or older?

- line 107: did you exclude some interventions? otherwise is not necessary to specify

- line 111-113: is not clear the frailty assessment

7. PLOS authors have the option to publish the peer review history of their article (what does this mean?). If published, this will include your full peer review and any attached files.

Reviewer #1: **Yes: **Blanca Martinez Lopez de Arroyabe

Reviewer #2: No

---

## [Decision Letter · Decision Letter 1]

5 Jan 2022

Influence of preoperative frailty on quality of life after cardiac surgery: protocol for a systematic review and meta-analysis

PONE-D-21-10086R1

Dear Dr. Fehlmann,

We’re pleased to inform you that your manuscript has been judged scientifically suitable for publication and will be formally accepted for publication once it meets all outstanding technical requirements.

Kind regards,

Laura Pasin

Academic Editor

PLOS ONE

Additional Editor Comments (optional):

Reviewers' comments:

Reviewer's Responses to Questions

**Comments to the Author**

1. Does the manuscript provide a valid rationale for the proposed study, with clearly identified and justified research questions?

Reviewer #2: Yes

2. Is the protocol technically sound and planned in a manner that will lead to a meaningful outcome and allow testing the stated hypotheses?

Reviewer #2: Yes

3. Is the methodology feasible and described in sufficient detail to allow the work to be replicable?

Reviewer #2: Yes

4. Have the authors described where all data underlying the findings will be made available when the study is complete?

Reviewer #2: Yes

5. Is the manuscript presented in an intelligible fashion and written in standard English?

Reviewer #2: Yes

6. Review Comments to the Author

You may also provide optional suggestions and comments to authors that they might find helpful in planning their study.

Reviewer #2: We thanks the authors for their reply. They ameliorated the manuscript and now its easier to read and it is clear. The argument is very interesting.

7. PLOS authors have the option to publish the peer review history of their article (what does this mean?). If published, this will include your full peer review and any attached files.

Reviewer #2: No

---

## [Editor Report · Acceptance letter]

25 Jan 2022

PONE-D-21-10086R1 

Influence of preoperative frailty on quality of life after cardiac surgery: protocol for a systematic review and meta-analysis 

Dear Dr. Fehlmann:

I'm pleased to inform you that your manuscript has been deemed suitable for publication in PLOS ONE. Congratulations! Your manuscript is now with our production department. 

Kind regards, 

on behalf of

Dr. Laura Pasin 

Academic Editor

PLOS ONE